# Surgical Significance of Berry’s Posterolateral Ligament and Frequency of Recurrent Laryngeal Nerve Injury into the Last 2 cm of Its Caudal Extralaryngeal Part(P1) during Thyroidectomy

**DOI:** 10.3390/medicina58060755

**Published:** 2022-06-01

**Authors:** Stylianos Mantalovas, Konstantinos Sapalidis, Vasiliki Manaki, Vasiliki Magra, Styliani Laskou, Stelian Pantea, Vasileios Lagopoulos, Isaak Kesisoglou

**Affiliations:** 1Third Department of General Surgery, Medical School, AHEPA University Hospital, Aristotle University of Thessaloniki, 54621 Thessaloniki, Greece; steliomantalobas@yahoo.gr (S.M.); vassiamanaki@gmail.com (V.M.); valia.magra@gmail.com (V.M.); stelaskou@gmail.com (S.L.); vaslag@gmail.com (V.L.); ikesis@hotmail.com (I.K.); 2Surgery Clinic II, University of Medicine and Pharmacy “Victor Babes”, 300041 Timisoara, Romania; panteastelian7@gmail.com

**Keywords:** thyroidectomy, Berry’s ligament, neurostimulation, recurrent laryngeal nerve, thyroid

## Abstract

*Background and Objectives*: Recurrent laryngeal nerve injury is one of the major complications of thyroidectomy, with the lateral thyroid ligament (Berry’s ligament) being the most frequent site of nerve injury. Neuromonitoring during thyroidectomy revealed three possible anatomical regions of the recurrent laryngeal nerve P1, P2, and P3. P1 represents the recurrent laryngeal nerve’s caudal extralaryngeal part and is primarily associated with Berry’s ligament. The aim of this systematic review is to identify the anatomical region with the highest risk of injury of the recurrent laryngeal nerve (detected via neuromonitoring) during thyroidectomy and to demonstrate the significance of Berry’s ligament as an anatomical structure for the perioperative recognition and protection of the nerve. *Materials and Methods*: This study conducts a systematic review of the literature and adheres to all PRISMA system criteria as well as recommendations for systematic anatomical reviews. Three search engines (PubMed, Scopus, Cochrane) were used, and 18 out of 464 studies from 2003–2018 were finally included in this meta-analysis. All statistical data analyses were performed via SPSS 25 and Microsoft Office XL software. *Results*: 9191 nerves at risk were identified. In 75% of cases, the recurrent laryngeal nerve is located superficially to the ligament. In 71% of reported cases, the injury occurred in the P1 area, while the P3 zone (below the location where the nerve crosses the inferior thyroid artery) had the lowest risk of injury. Data from P1, P2, and P3 do not present significant heterogeneity. *Conclusions:* Berry’s ligament constitutes a reliable anatomical structure for recognizing and preserving recurrent laryngeal nerves. P1 is the anatomical area with the greatest risk of recurrent laryngeal nerve damage during thyroidectomy, compared to P2 and P3.

## 1. Introduction

Recurrent laryngeal nerve injury represents one of the most severe complications of thyroidectomy. According to the American Society of Head and Neck Surgery, voice changes after thyroidectomy are believed to range from 30–87%, whereas recurrent laryngeal nerve damage is underrated (3–5%) and appears to be approaching 10% [1,2,3]. There are multiple probable locations along the path of the recurrent laryngeal nerve that can be damaged during thyroidectomy. Sir James Berry, in one of his references in 1888 in the Journal of Anatomy community of Great Britain and Ireland, described in detail the posterolateral ligament of the thyroid gland characterizing it as a cremaster ligament. He observed that the rates of recurrent laryngeal nerve injuries were increased in that particular anatomic region [4]. Since then, many writers have described the increased probability of injury of the recurrent laryngeal nerve in the pre-mentioned site [5,6,7,8,9,10]. However, this was widely approved with the systematic use of intraoperative neurostimulation during thyroidectomy and its application in several operations on the head and neck [11,12,13,14,15,16,17,18,19,20,21,22,23,24,25,26]. Prior to the use of intraoperative neurostimulation, the only approved assessment for recurrent laryngeal nerve injury was postoperative laryngoscopy and vocal cord examination, at least when the situation slipped the surgical team’s visual perception intraoperatively. The loss of signal during recurrent laryngeal nerve stimulation offered the potential to identify nerve damage and also to indicate the specific moment and place of injury. Initially, Marcus [11] and then Chiang and Snyder [27,28], using this technique, showed that Berry’s ligament is the most frequent region of recurrent laryngeal nerve injury after comparing this site to other regions of nerve’s course. More specifically, they mentioned that the posterolateral ligament is the spot where injury of the nerve happens at a rate of more than 75% of the whole injury incidents due to the traction force upon the nerve [27,28]. The recurrent laryngeal nerve has several different divisions referring to its intracervical course. The most popular is the one occurring after the use of neurostimulation. According to this classification, the recurrent laryngeal nerve is divided into three zones. P1 is the region beginning where the nerve crosses the inferior thyroid artery and ending at its entrance into the larynx. P2 is the region surrounding the former one, and P3 is the region below the pre-mentioned cross. P1 is the region referring to the last 2 cm of the caudal extralaryngeal part of the recurrent laryngeal nerve and is clinically significant. Berry’s ligament is substantially related to the last region. This happens because it runs a large length of that distance, they have a common embryological origin, and they are both covered with loose connective tissue [4,29]. For that reason, the American Society of Head and Neck Surgery refers to the ligament with the term “anatomical complex of Berry’s ligament” and not simply as ligament [1]. This review and meta-analysis aim to investigate the frequency of recurrent laryngeal nerve injuries in the anatomical regions P1, P2, and P3 based on intraoperative loss of signal. Analysis of all the existing articles may enhance the statistical power and come to a more solid conclusion on the importance of Berry’s ligament as the anatomical spot for the recognition and protection of the recurrent laryngeal nerve.

## 2. Materials and Methods

The analysis was carried out by following the PRISMA guidelines for systematic review and meta-analyses [30], but also the guidelines of Henry et al. (2016) for anatomic systemic reviews [31]. The research also has the features of meta-analysis because it is mostly an anatomical study, and there are no Randomized Control Trials (RCTs) in this specific era. The research was registered to PROSPERO (ID 336474)

### 2.1. Surgical Techniques and Neurostimulation Procedure

The studies included patients who had undergone thyroidectomy with or without central lymph node dissection. The case of additional lateral neck lymph node dissection was not considered due to the fact that recurrent laryngeal nerve is found in the central cervical compartment and is usually injured only in the central lymph node dissection. The research also includes all thyroidectomy surgeries, including endoscopic or camera-assisted procedures. As far as neurostimulation is concerned, it can be simple or continuous. The neurostimulation system used, even whether it follows standard practice (L1—V1—R1—R2—V2—L2), is not taken into account. According to the data of each study group, we characterize every loss of signal type I (LOS I) as intraoperative recurrent laryngeal nerve injury. Based on the current literature and international data, loss of signal is considered to be the loss of audio tone and/or the decrement of the amplitude of the nerve below 100 μV after stimulation of 1–2 mA, noted in the electromyography that is being recorded. In the pre-mentioned results, false positive records of loss of signal are not taken into consideration (problems of electrodes etc.). Furthermore, neither the probability of postoperative temporal or permanent harm nor the type of injury is investigated (incision, thermal injury, crush injury, or increased forces applied on the nerve). Regardless of thyroidectomy techniques or neurostimulation methods, our main goal was to include studies with pure and clear quantitative results when it comes to areas P1, P2, and P3 and their relationships with nerve injury. Where this was not achievable, data were quantified based on the descriptions and schemes of writers in order to classify them into the categories P1, P2, and P3 in an explicit and precise manner.

### 2.2. Search Strategy

The analysis was carried out by following the PRISMA guidelines for systematic review and meta-analyses.

The search string that we used is the following: ((((IONM OR intraoperative neuromonitoring OR nerve monitor*) AND (recurrent laryngeal OR inferior laryngeal)) AND injury)) AND (thyroidectomy OR “thyroid surgery”))).

The search engines used for this study were PubMed, Scopus, and Cochrane. We will also check the reference list of the articles for references that may have been missed by electronic search. An effort was made to correct the overlap between the three search engines. Among 711 detected studies, 402 derived from PubMed, 306 from Scopus, and 3 from Cochrane. By removing the overlapped/double studies that were 247, a total 464 studies were used for the purpose of this study (Figure 1).

### 2.3. Criteria for Selection of Studies

Data extraction and assessment of the quality of the studies were performed by two independent investigators and, in case of disagreement, by a third researcher. All researchers used the program Adobe Acrobat Reader (Adobe Inc., Mountain View, CA, USA).

For the selection of studies, two searches were conducted in a row. Studies whose subject was inappropriate were excluded in the first search, which included just reviews. In the second one, the whole text was read. Regarding inclusion criteria, the following keywords (1) “Berry”, (2) “P1” or “P2” or “P3”, (3) “distal 2 cm” or “extralaryngeal” were used. All fully developed articles that were candidates for inclusion in the meta-analysis were thoroughly read. The major criteria for including studies in this meta-analysis were clear quantitative and sufficient data for the purpose of this research, and more precisely quantitative results referring to the main anatomical regions P1, P2, and P3. Criteria of exclusion were (1) the non-quantitative or clear results, (2) case reports or case series, (3) letter to the editor, (4) other meta-analyses or systematic reviews articles, (5) animal studies-researches, (6) studies with irrelevant data, (7) congress presentations, and (8) studies with overlapped data. At least two researchers-readers assessed each study’s suitability for inclusion in the meta-analysis. In cases where quantitative results were unable to fit into the categorization system P1, P2, and P3 (e.g., anatomical images or categorization based on the centimeters of the injury from the entrance spot of the nerve into the larynx), an effort was made to place them in the aforementioned system with the consensus of the meta-analysis researchers. 

### 2.4. Data Extraction 

Data of each study were classified by year, country, type of study/research (prospective, retrospective), total number of nerves at risk and data concerning injury in regions P1, P2, and P3 from tables and diagrams/charts funnel plots and forest plots). Subgroup analysis was also applied in some variables.

### 2.5. Statistical Analysis

All data statistical analyses were performed via IBM SPSS Statistics for Windows, Version 25.0. (IBM Corp., Armonk, NY, USA) and Microsoft Office XL software (Microsoft Corporation, One Microsoft Way, Redmond, WA, USA).

Results with *p* < 0.05 were considered as statistically significant. Normality of the variables was checked by Shapiro–Wilk test. Mean and median value was calculated for every variable. For the non-normal distributed data, Friedman test was used. Statistical analysis between two groups was obtained through paired samples *t*-test or Wilcoxon test (for non-normally distributed data). Statistical heterogeneity was assessed by using Cochrane’s Q and I^2^ tests. For the evaluation of Cochrane’s Q test, *p* < 0.05 was considered as statistically significant. I^2^ test results were interpreted by the following classification: (1) 0–40% there is not significant heterogeneity, (2) 30–60% might suggest moderate heterogeneity, (3) 50–90% might indicate significant heterogeneity, and 75–100% could represent significant heterogeneity. If *p* < 0.1 and I^2^ > 50% (apparent heterogeneity) a random-effects model was applied, otherwise a fixed-effects model was chosen. Subgroup analysis was performed for the analysis of ratios of injury in different anatomic regions (P1, P2, and P3). 

## 3. Results

Eighteen studies were finally evaluated [11,12,13,14,16,17,18,20,21,22,23,24,25,26,28,32,33]. A total of 9191 nerves at risk (NAR) with the use of neurostimulation were identified. In 277 cases loss of signal type I occurred. Final results are represented in Table 1. Table 2 represents the mean value and standard deviation of ratios in each study referring to the spots where most of injuries of recurrent laryngeal nerve happened during thyroidectomy. According to the research, injuries occurred in region P1 at a rate of 70.88%, in region P2 at a rate of 20.66%, and in region P3 at a rate of 8.47%. (Table 2).

Due to the non-normal distribution from the statistical Shapiro–Wilk test, the non-parametric Friedman test was applied to assess the dependence of variables “region P1”, “region P2”, and “region P3”. More specifically, the null hypothesis is that the mean values of the variables are equal, and the alternative one (alternative hypothesis) is that there is a statistically significant difference between them. Null hypothesis was rejected (F (2) = 25.569, *p* = 0.00, <0.05). The mean value in region P1 (2.86) is higher/larger than that of region P2 (1.86). Similarly, the mean value in region P2 (1.86) is higher/larger than that of region P3 (1.28) (Table 2).

Paired samples *t*-test between regions P1 and P2 was performed and the mean value of injury risk in region P1 (mean = 70.88) was statistically higher (t (17) = 5635, *p* = 0.000, <0.05) than in region P2 (mean = 20.66).

Wilcoxon test between regions P3, P2 was performed. This particular test was used due to the inexistence of normal distribution in region P3. The mean value in region P3 (1.89) was statistically lower (Z = −2528, *p* = 0.011, <0.05) than in region P2 (mean = 3.50).

In region P1, loss of signal was observed in 180 nerves. Region P1 was also observed in 66% (CI 95%: 57–74%) of cases. Ιn region P2 of the nerve, a loss of signal was observed in *n* = 63 nerves corresponding to a 24% (CI 95%: 18–30%) ratio, while in region P3, it was observed in *n* = 34 corresponding o 12% (CI 95%: 57–74%). Publication bias and heterogeneity concerning P1, P2, and P3 value was also assessed via funnel plots.

It is also important to notice that the results of studies for region P1 have moderate heterogeneity (I^2^ = 47%, *p* = 0.01) (Figure 2, Figure 3 and Figure 4) while in regions P2 (Figure 4 and Figure 5) and P3 (Figure 6 and Figure 7) there is no heterogeneity (I^2^ = 32%, *p* = 0.10 and I^2^ = 5%, *p* = 0.40 respectively). In each case all the results are considered to be normally distributed.

Due to the fact that in some studies, new techniques of thyroidectomy (endoscopic andvideo-assisted) were performed, and in others, central lymph node dissection took place, these studies were excluded (Figure 8, Figure 9 and Figure 10). Based on new data, it was found that recurrent laryngeal nerve injury is more frequent in the anatomic region P1 in a ratio of 69% (CI 95%: 61–76%), in region P2 in 20% (CI 95%: 15–26%) and in P3 in 12% (CI 95%: 6–17%). Researches were also found to be more normally distributed between themselves, as for the region P1 I^2^ = 30%, for region P2 I^2^ = 0% and for region P3 I^2^ = 0%.

Finally, the results were tested for the existence of a statistically significant difference between themselves.

In summarizing, the research question is the presence or not of a statistically significant difference between the ratio of injured nerves with loss of signal (LOS I) in region P1 in relation to injured nerves with loss of signal in the P2 region. Similarly, a comparison between regions P1 and P3 and P2 and P3 was applied. Between regions P1 and P2 and also between P2 and P3, a statistically significant difference was observed (*p* < 0.05). The results of the research indicate that region P1 has higher ratios of injuries while those in region P3 are the lower ones. 

## 4. Discussion

Recurrent laryngeal nerve injury is one of the major complications of thyroidectomy, with the lateral thyroid ligament (Berry’s ligament) being the most frequent site of nerve injury. Post-thyroidectomy voice disorders usually approach a ratio of 10% [13,14]. Prior to the use of intraoperative neurostimulation, the only approved examination for recurrent laryngeal nerve injury was postoperative laryngoscopy and vocal cord assessment whenever the injury escaped intraoperative recognition. The ability to define the exact time and region of the injury intraoperatively was made possible by a loss of signal during stimulation of the recurrent laryngeal nerve [11,12,13,14,15,16,17,18,19,20,21,22,23,24,25,26]. According to Snyder et al. (2007), injured nerves are usually intact intraoperatively (0.45%) [12]. This fact has been attributed to a functional type of injury of the nerve during thyroidectomy. Chiang et al (2008) mentioned that functional injuries of the nerve constitute over 75% of all injuries. This sort of damage occurs as a result of traction pressures acting on Berry’s posterolateral ligament during thyroid gland central rotation [27]. Despite the fact that this ligament was initially described by Gruber and Henly in1880 [5], its surgical importance was a point of study for Sir James Berry in1888, and that is why the ligament bears his name [4]. Sir James Berry was the first to observe that the recurrent laryngeal nerve is more vulnerable to injury in that specific location. Many researchers who came after him, particularly after the widespread use of intraoperative neurostimulation in head and neck surgery, described this relation with clarity [11,12,13,14,15,17,18,19,20,21,22,23,24,25,26,26,27,28]. 

According to the pre-mentioned results of the study, 2/3 (66%) of nerve injuries that are detected with intraoperative neurostimulation are located in the last caudal 2 cm of the extralaryngeal part of the nerve (P1 region). A total of 24% of the injuries are observed in the region surrounding the inferior thyroid artery, and only 12% are located in the region under the inferior thyroid artery. The dominance of injuries in P1 was proven to be statistically significant when compared to P2 and P3 individually.

This is owing to the unique anatomic features of the P1 area, specifically due to Berry’s ligament. First of all, Berry’s ligament extends from the inferior thyroid artery to the recurrent laryngeal nerve’s entry into the larynx. According to Sasou et al. [4], the ligament may extend from the cricoid cartilage to the third tracheal cartilage, but Thompson et al. believe it may extend until the fourth tracheal cartilage [9]. Regardless of the nerve’s relationship to the ligament (anterior, lateral, etc.), both of these structures are coated by loose connective tissue. Berlin et al. since 1933 characterized this region as an attached/unified zone [6,7], while Salama and McGrath (1992), Sasou (1998), and Seprell (2010) certify the pre-mentioned fact [4,29,34]. However, there is a link even from an embryological standpoint. Cricoid cartilage, where the thyroid gland will be hooked, is derived from the mesenchyme of the fourth and sixth pharyngeal arches, and the recurrent laryngeal nerve is the nerve of the sixth pharyngeal arch. In addition, the inferior laryngeal artery, as a terminal branch of the inferior thyroid artery (fourth aortic arch) in adult life, passes through Berry’s ligament at its inferior border [35,36]. For this reason, the American Society of Head and Neck Surgery refers to the structure as an anatomical complex rather than just as a ligament [1].

There are several differing opinions on whether or not the recurrent laryngeal nerve perforates Berry’s ligament. In the meta-analysis of Henry et al. (2017), it occurs in 7% of cases [37], while in the cadaveric research of Yalcin and Ozan (2006), such cases are recorded in great detail [38]. On the other hand, Sasou (1998) and Cakir et al. (2006) verify this relation [4,39]. According to Seprell et al. (2010), this disparity appears to be a terminology issue [29]. For this purpose, he proposed the theory of two levels [29]. According to that idea, the ligament is defined by two layers: one superficial vascular layer that, once dissected, reveals the recurrent laryngeal nerve in its final path, and one true ligament, Berry’s ligament, which constitutes the second deep component of the entire structure. However, the thyroidectomy procedure he suggests is the standard one, which discloses the nerve intraoperatively. In each scenario, including the one where the nerve perforates the Berry ligament, the majority of injuries occur in this particular region.

Berry’s ligament is a stable anatomical feature associated with the recurrent laryngeal nerve (Figure 11, Figure 12 and Figure 13). In both meta-analyses of Henry et al. (2016 and 2017), it was discovered to be the most accurate structure since the nerve is superficially related to the ligament in 78% of cases [37]. In 74% of cases, the recurrent laryngeal nerve passes posteriorly to Zuckerkandl’s lobe. The tracheo-oesophageal groove is the next clue, with the recurrent laryngeal nerve appearing to lie within the groove in 64% of cases, and the most unstable relationship is with the inferior thyroid artery, where the nerve tends to pass posterior in 58% of cases. As a result, the consistent relationship with Berry’s ligament defines this hint as the most essential one for recognizing the recurrent laryngeal nerve. We believe that Pelizzo’s (1998) description of Zuckerkandl’s lobe as an arrow pointing to the recurrent laryngeal nerve is a fact that involves the Berry’s ligament equally and accurately [40]. According to Seprell et al., in addition to Zuckerkandl’s lobe, the recurrent laryngeal nerve is positioned directly underneath the thyroid’s vascular layer. The following procedures must be carried out during a lateral access thyroidectomy: First, the vascular layer must be penetrated. Second, the genuine Berry’s ligament must be identified, which, as previously stated, has the most stable relationship with the recurrent laryngeal nerve. Third, the recurrent laryngeal nerve must be visualized and identified, primarily near the region where Berry’s ligament fibers approach the trachea. Fourth, the imaginable line of the nerve’s path with regard to the cricothyroid membrane and the tracheo-oesophageal groove, particularly at the level inferior to the inferior thyroid artery, must be evaluated. This hypothetical course is estimated in a manner similar to the study provided by Shindo et al. (2005), and if possible, the nerve degrees in respect to the tracheo-oesophageal groove are calculated [41]. As previously stated, recurrent laryngeal nerve injuries in the P3 region approach a ratio of 12%, indicating a statistically significant difference in comparison to the region of Berry’s ligament, and the studies in our meta-analysis verifying this fact do not show significant heterogeneity (I^2^ = 5%). This is true even if we include studies in which advanced thyroidectomy techniques were used. As a result, the primary focus must be on identifying Berry’s ligament, which will lead to the identification of the recurrent laryngeal nerve.

The point of the nerve that retains a closer position to the median line of the trachea, which is commonly referred to as the safety point in thyroidectomy, has been discovered to be the location of the nerve’s entrance into the larynx [42]. This relation is maintained when central access is preferred due to the transportation of traction forces laterally [27]. This technique (central access) is preferred in large nodules or in endoscopic thyroidectomy. However, according to the American Society of Head and Neck Surgery^1^ guidelines, lateral access remains the primary choice in thyroidectomy. The previously indicated relationship is disrupted in situ during lateral access. As a result, the closest point of the nerve to the tracheal midline may not be the point of entry into the larynx but rather a position more peripheral, likely into Berry’s ligament. The traction force provoked by frontal and central rotation of the thyroid gland not only harms the nerve due to traction but also creates an artificial genu in the recurrent laryngeal nerve [27]. Because it does not deviate considerably from the median cervical line, this genu is more prone to direct nerve injury. Furthermore, the last few millimeters of the recurrent laryngeal nerve are covered by fibers of the cricopharyngeus muscle in two-thirds of cases, indicating that there is an extra anatomical component that provides protection in the recurrent laryngeal nerve at its entrance into the larynx [43]. More research on the anatomical region of the nerve’s caudal 2 cm in relation to its injury is required. We believe that this anatomical region should be divided into three distinct zones, each of which should be associated with the likelihood of recurrent laryngeal nerve injury. The first zone extends from the nerve’s crossing with the inferior thyroid artery to the lower attachment of Berry’s ligament to the trachea, the second until the cricoid cartilage-cricopharyngeus muscle, and the last between the inferior border of the cricopharyngeus muscle and the point of the nerve’s entrance into the larynx (Figure 12 and Figure 13). Berry’s ligament is the most reliable anatomical structure in the process of recurrent laryngeal nerve recognition, and its surgical significance is attributed to the fact that the ligament’s region is the one where the recurrent laryngeal nerve is prone to most perioperative injuries in the domain of thyroid and parathyroid surgery. This knowledge has been known since Sir J. Berry’s time (1888). This systematic review not only confirms the pre-mentioned reference but also improves the quality of evidence concerning perioperative neurostimulation in the field of head and neck surgery.

## 5. Conclusions


Berry’s ligament is a stable anatomical structure in relation to the recurrent laryngeal nerve because it has the most consistent link with the nerve (in 75% of cases, the nerve passes superficially to the ligament).Berry’s ligament is surgically significant since the majority of recurrent laryngeal nerve injuries (functional or non-functional) are detected in that anatomical location (66%).This systematic review and meta-analysis provide a higher quality of evidence in the aforementioned anatomical relationship. It is the first meta-analysis in this era referring to the anatomical regions where recurrent laryngeal nerve injuries are observed (P1, P2, and P3) after the widespread use of intraoperative neurostimulation in head and neck surgery.


## Figures and Tables

**Figure 1 medicina-58-00755-f001:**
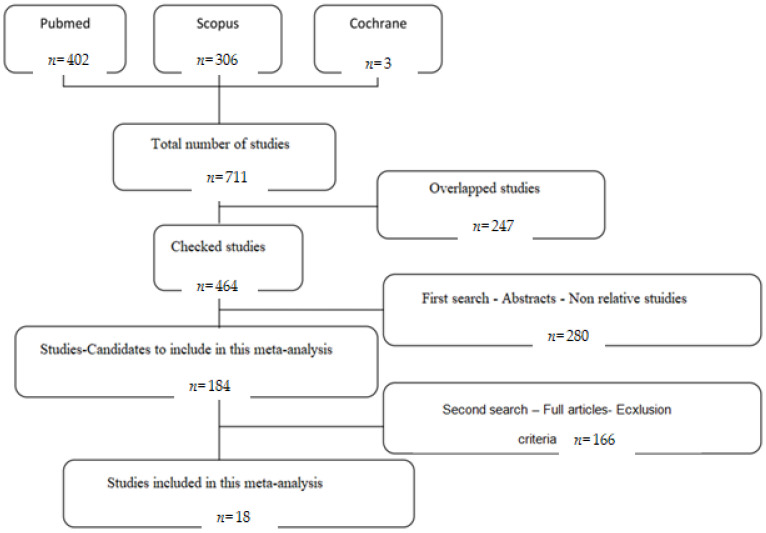
Flow chart.

**Figure 2 medicina-58-00755-f002:**
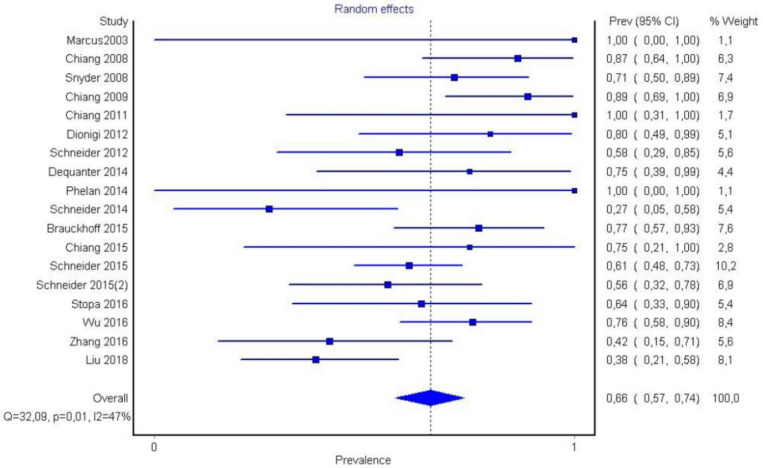
Forest plot P1.

**Figure 3 medicina-58-00755-f003:**
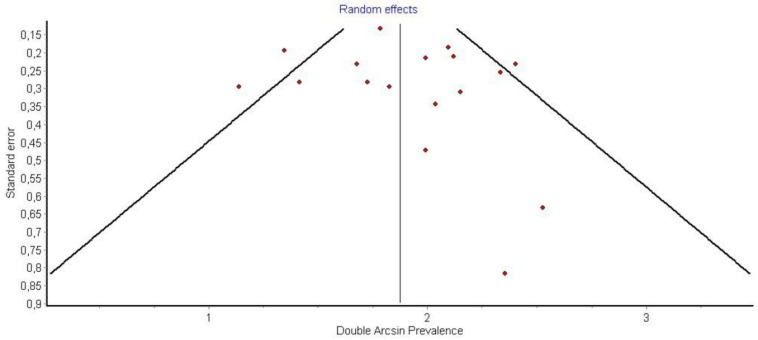
Funnel plot P1.

**Figure 4 medicina-58-00755-f004:**
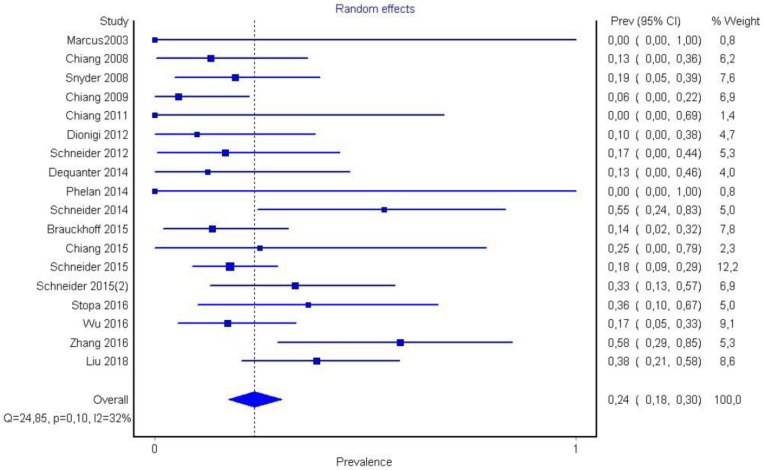
Forest plot P2.

**Figure 5 medicina-58-00755-f005:**
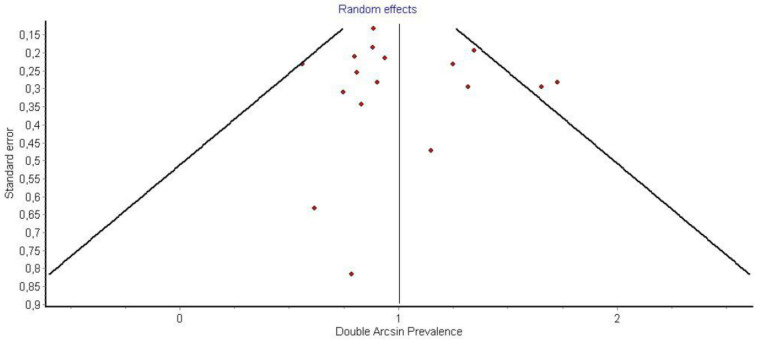
Funnel plot P2.

**Figure 6 medicina-58-00755-f006:**
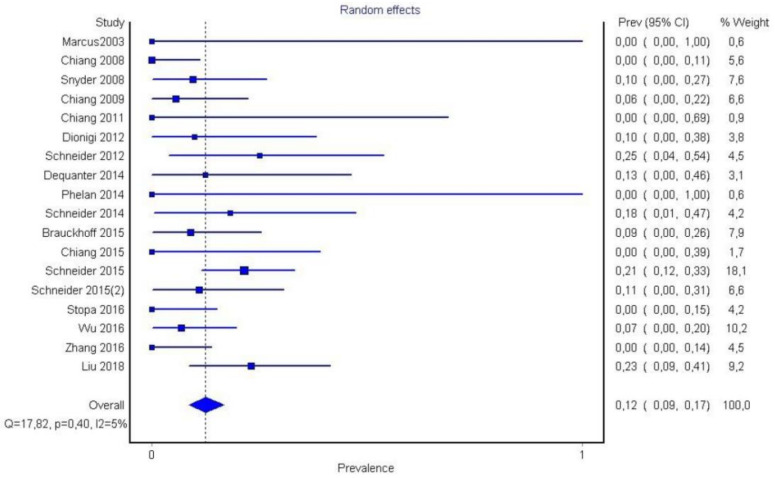
Forest plot P3.

**Figure 7 medicina-58-00755-f007:**
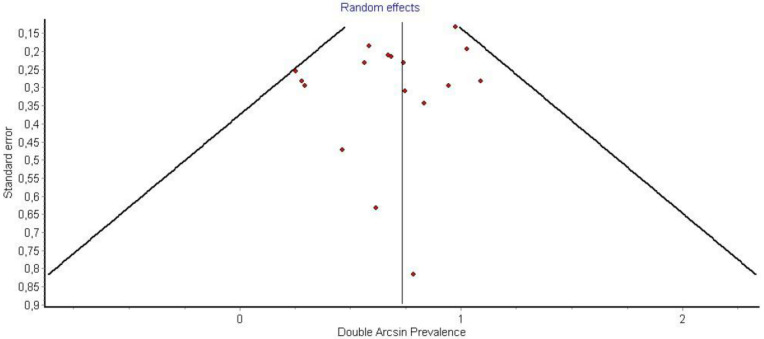
Funnel plot P3.

**Figure 8 medicina-58-00755-f008:**
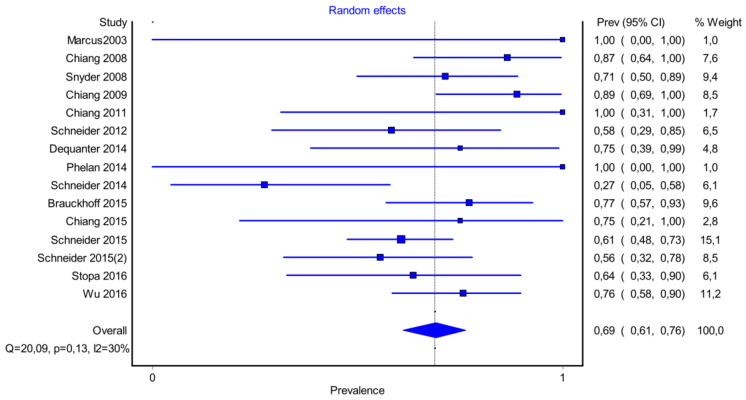
Forest plot P1 (subgroup analysis).

**Figure 9 medicina-58-00755-f009:**
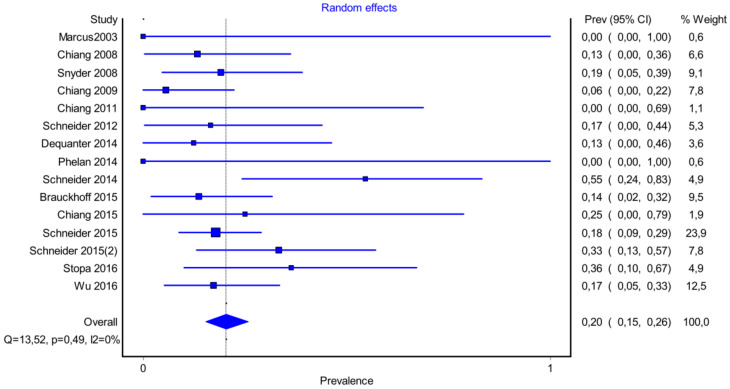
Forest plot P2 (subgroup analysis).

**Figure 10 medicina-58-00755-f010:**
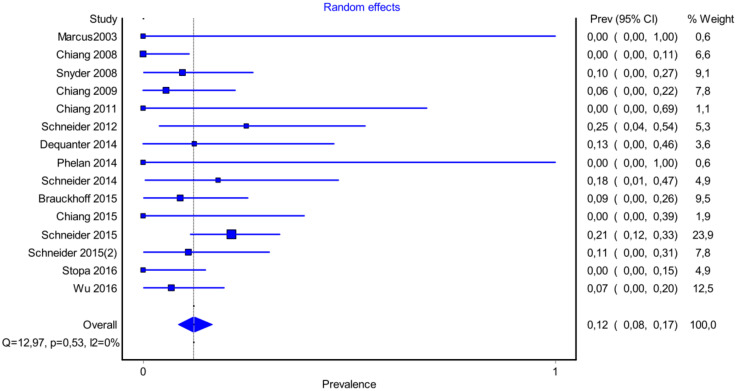
Forest plot P3 (subgroup analysis).

**Figure 11 medicina-58-00755-f011:**
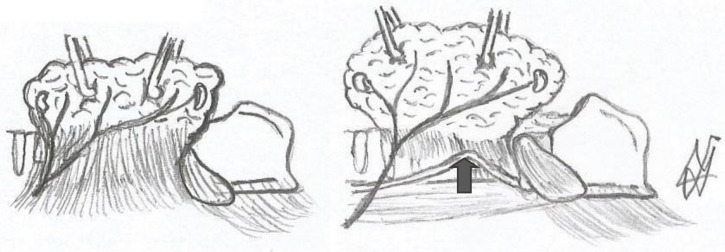
Thyroid, Berry’s ligament and the recurrent laryngeal nerve (drawing of the authors). The arrows show the nerve.

**Figure 12 medicina-58-00755-f012:**
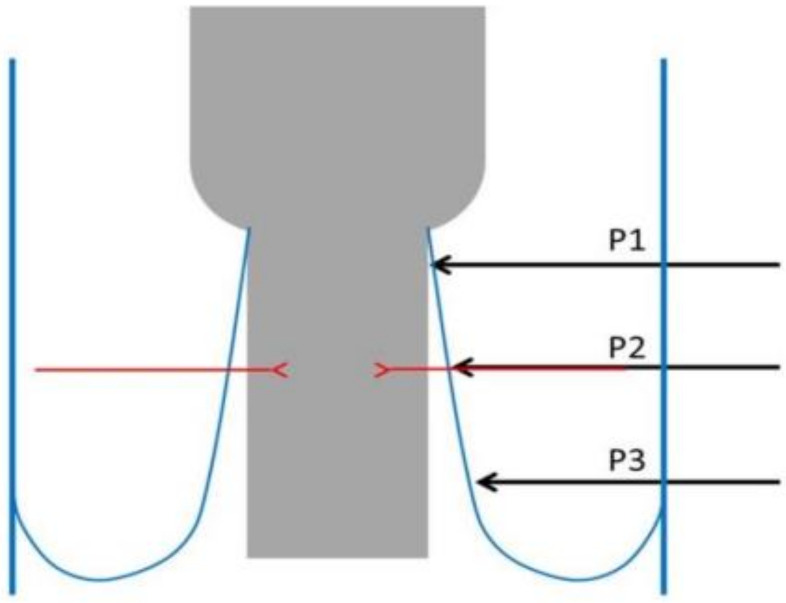
Anatomic regions where NRL nerve injuries of usually occur.

**Figure 13 medicina-58-00755-f013:**
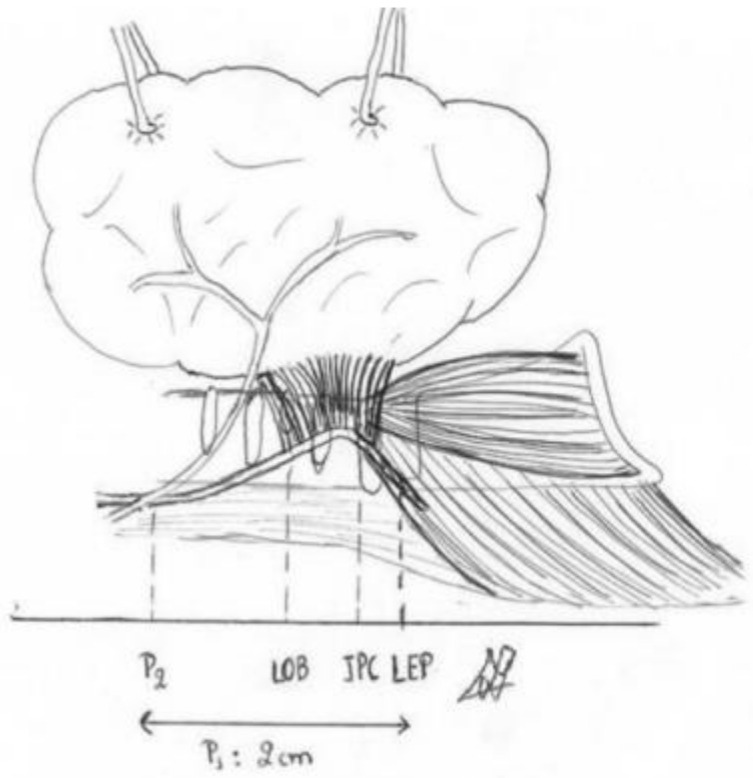
The picture shows P1 and P2 regions (painted by one of the authors).

**Table 1 medicina-58-00755-t001:** Data of meta-analysis.

	P1	P2	P3	Summa	Country	Study	NAR	Technique
Marcus (2003) [11]	1	0	0	1	USA	P	81	Classic
Snyder (2008) [12]	15	4	2	21	USA	P	666	Classic
Chiang (2009) [13]	16	1	1	18	Taiwan	P	435	Classic
Chiang (2011) [14]	2	0	0	2	Taiwan	P	506	Classic
Schneider (2012) [16]	7	2	3	12	Germany	R	52	Classic
Dequanter (2014) [17]	6	1	1	8	Belgium	P	175	Classic
Schneider (2014) [18]	3	6	2	11	Germany	R	2086	Classic
Brauckhoff (2015) [19]	17	3	2	22	Norway	P	87	Classic
Chiang (2015) [20]	3	1	0	4	Taiwan	P	168	Classic
Schneider (2015) [21]	34	10	12	56	Germany	P	115	Classic
Stopa (2016) [22]	7	4	0	11	Poland	P	1000	Classic
Wu (2016) [23]	22	5	2	29	Taiwan	P	522	Classic
Zhang (2017) [24]	5	7	0	12	China	P	156	Endo
Liu (2018) [25]	10	10	6	26	China	P	1273	CND
Phelan (2014) [26]	1	0	0	1	Germany	P	204	Classic
Chiang (2008) [27]	13	2	0	15	Taiwan	P	173	Classic
Schneider (2015) (2) [28]	10	6	2	18	Germany	R	1291	Classic
Dionigi (2012) [33]	8	1	1	10	Italy	P	201	VA

P: prospective; R: retrospective; VA: video-assisted; Endo: endoscopic; CND: central neck dissection; NAR: nerves at risk.

**Table 2 medicina-58-00755-t002:** Mean value and standard deviation of ratios in each study referring to the spots where most of injuries of recurrent laryngeal nerve happened during thyroidectomy.

Regions	Mean	S.D.
Region P1	70.88	21.22
Region P2	20.66	17.39
Region P3	8.47	8.71

## Data Availability

Not applicable.

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
