# Peer review of "Surgical Significance of Berry’s Posterolateral Ligament and Frequency of Recurrent Laryngeal Nerve Injury into the Last 2 cm of Its Caudal Extralaryngeal Part(P1) during Thyroidectomy"

_medicina, 2022, doi:10.3390/medicina58060755_

Round 1

Reviewer 1 Report

Respected Authors,
First and foremost, I would like to congratulate you all for your contribution to the scientific 
community. After reading this manuscript, I felt that this current version should be enhanced, 
so that the reader shall enjoy it while reading the manuscript. I am mentioning my remarks 
below.
1. Format consistency is missing. I wonder why or how none of the authors noticed it. 
Like shells on the beach, there are so many minor mistakes. Initially, I was making a 
note of it (lines no 15, 22, 35, 46, 55, 67, ….). But later, I lost track!
2. Future tense is used in some parts of the manuscript. Example: Line no 74 and 107.
3. Line 116 and 117. What happens if there is a disagreement by a third researcher?
4. How many researchers or investigators were involved in this study? Do they know 
Anatomy? What were their designations?
5. What is the significance of lines 117 and 118?
6. Line 119 and 120. Who determines whether the study is appropriate or not?
7. Line 135 and 136. German to English. Why this line is specifically mentioned? What is 
the importance of it?
8. Line 158. 18 studies were finally evaluated [91-106]. 91 to 106 is 16 studies. Where 
are the remaining 2?
9. Out of 464 studies, only 18 were included in this study. That’s 3.87%. How do you 
justify it? Was the search done wrong? Were the keywords appropriate?
10. Why were only 3 keywords used? A simple search of “Laryngeal Nerve Injury” in 
ScienceDirect reveals several thousands of papers in regards to Thyroidectomy.
11. Was the Literature Review done properly?
12. What is the need for this study? Why this study was conducted? Mention the reasons.
13. Under the Results section, in several places, P-value is mentioned using a comma ‘,’ 
instead of ‘.’
14. A simple diagrammatic representation of the recurrent laryngeal nerve with P1, P2, 
and P3 can help the readers to understand its anatomical location of it.
15. Instead of posting an image of the data (Panel 1), an actual table could have been 
provided.
16. In figure 12, parts are not marked. Which is a ligament? Which is the nerve? Readers 
can mistake between blood vessels and nerves if it is not explained.
17. The data mentioned in the supplementary file does not match the manuscript. 
Example: Line numbers, figure numbers

Author Response

Dear reviewer, 

Thank you very much for your time and your useful comments concerning our study. Below I will try to respond to your comments:

  1. We tried to correct as much format mistakes as possible
  2. We corrected the mistake
  3. The third researcher is the Professor of our department and very experienced endocrine surgeon. We respect his opinion and if for a precise reason there was a disagreement, for example about to include or not a study or to classify the described injuries in P1 P2 and P3, the team followed professor's advice so as not to have bias.
  4. The researchers were all surgeons. The team consisted of a general surgeon, general surgery resident and the Professor of the department (third).  All three researchers are aware of anatomy.
  5. "Data extraction and assessment of the quality of the studies was performed by two independent investigators and in case of disagreement by a third researcher. All researchers used the program Adobe Acrobat Reader" was mentioned in order to highlight that more than one researcher checked the literature in order not to miss articles that could be included in the study
  6. At first, the two main investigators separetely checked the literature the way we describe in the manuscript. If there was a disagreement between them, then the third researcher was the one to decide if a study should be included or not according to the inclusion criteria mentioned in the test.
  7. We deleted those sentences as they do not offer a lot to the study
  8. A mistake was made. The missing references were added and We corrected the reference list via Zotero
  9. We do not think that something went wrong. We believe there is publication bias and there is little reference in the literature in injured nerves intraoperatively. In our study we chose studies that described any kind injury of the nerve.
  10. Our search string was : ((((IONM OR intraoperative neuromonitoring OR nerve monitor*) AND (recurrent laryngeal OR inferior laryngeal)) AND injury)) AND (thyroidectomy OR "thyroid surgery"))) and the flow chart describes the deduction of studies
  11. We believe that the research methodology of our study was organised and done properly
  12. This study is the first meta analysis to statistically prove that the most usual injury of the NRL nerve occurs in P1 zone near Berry's ligament and this is of great significance during thyroidectomy. This is of great significance among endocrine surgeons. This study adds to the literature as it offers evidence based proof for the injury of the nerves.
  13. We corrected it
  14. We added two figures (13,14)
  15. We added as picture because it suits better to the sample document and we can modify the size
  16. We added an arrow showing the nerve
  17. We will update the supplementary data when we have made all the changes suggested by the reviewers

Dear reviewer thank you so much for your time. Some of your suggestions helped us a lot figuring out some mistakes that should have been corrected. We are looking forward to hearing your opinion concerning these alterations. 

Reviewer 2 Report

Injury to the recurrent laryngeal nerve (RLN) is a dreadful complication of thyroid and parathyroid procedures. In this Paper, Mantalovas et al. present their data from a review of 18 papers from a 15 Year period (2003 until 2017) retrieved from the Pubmed, Scopus and Cochrane databases. They attempt to identify the most likely region for injury to the RLN as determined by intraoperative neuromonitoring locating the site of a loss of signal.

Using techniques also applied in meta-analyses, they find the region above the intersection with the artery and close to the entry site of the RLN into the larynx to be the most likely site of injury.

The paper is reasonably well written, the methods are generally well described, although it is significant that there is a lack a grading system to assess the quality of the papers included (e.g. : Dijkers M. Introducing GRADE: a systematic approach to rating evidence in systematic reviews and to guideline development. Knowl Translat Update. 2013;1:1–9.; or: Higgins JP, Altman DG, Sterne JA. Chapter 8: Assessing the risk of bias in included studies. In: Cochrane Handbook for Systematic Reviews of Interventions: The Cochrane Collaboration 2011. updated 2017 Jun, available from http://handbook.cochrane.org.)).

If data on grading is available, it should be presented, because if this analysis includes (too many) low-quality studies, its results will be biased and incorrect

There are, some specific concerns that I would like to raise:

- Search engines: only thee search engines were used – please explain

- Search period: only studies from 2003 to 2015, a 15 year period, were searched – why was this specific period chosen ? Using the very search string the authors have used, more than 300 papers have been published since 2015. - please explain in detail.

- Meta-analytical operational construct: only at a later stage do the authors “exclude” studies that employ special surgical techniques (endoscopic, video assisted) or thyroid pathology (thyroid cancer cases with more “extensive” surgeries) – these data are presented as a “subgroup” analysis.

In fact, it would appear, that the majority of cases and studies available for this analysis would report from “conventional” thyroid surgeries in non-cancer patients. From a standpoint of operational homogeneity of the studies included, it would therefore be only appropriate, that those studies involving cancer cases as well as endoscopic procedures should have been excluded in the first place - because they are subgroups themselves – please explain. Also, the funnel plots or any other measure of heterogeneity for this "subgroup analysis" are missing.

Figure 2.: the format is unusual and not commonly used in the scientific literature – please revise this graph

Finally, this paper does require major English editing. There are repetitions of larger bodies of text both in the introduction as well as the discussion, the results section is very difficult to read – and the somewhat blurry description of the P1, P2 and P3 classification might better be accompanied by a graph – referring to the many graphs and descriptions the international neuromonitoring group has presented. The discussion contains a lot of surgical narrative advice – that was not the scope of this paper nor does it belong here and avoiding these narrative passages should also significantly reduce the word count.

Author Response

Dear reviewer,

Thank you very much for your time and your useful comments. Below I will try to respond to your comments:

  • Data on grading are not availale
  • We had access in three searching machines Pubmed, Scopus and Cochrane and we believe that they offer an adequate number of studies. Moreover, as far as research methodology is concerned in systematic reviews and meta-analysis, using three searching machines is quite efficient as they cover a wide range of good published studies.
  • The meta analysis is based in articles until 2020 as this study was the Msc thesis of the first author. The team of the authors also searched for articles during 2021 but due to the inclusion criteria these studies were not chosen as some of them did not describe the region of injury, used neuromonitoring without indicating the exact region of injury or not precisely describing the region of injury so as to classify it in one of the three groups (P1, P2, P3)
  • We did not exclude these studies with different techniques because our aim was to include studies that mention the site of injury in the text or in the figures without taking into consideration the technique
  • We changed Figure 2
  • We also added two figures explaining the regions P1, P2, P3

Thank you very much for your comments. We are looking forward to hearing your opinion on these alterations.

Reviewer 3 Report

No comments.

Author Response

Thank you very much!

This manuscript is a resubmission of an earlier submission. The following is a list of the peer review reports and author responses from that submission.